# *Mycobacterium tuberculosis* Transmission in High-Incidence Settings—New Paradigms and Insights

**DOI:** 10.3390/pathogens11111228

**Published:** 2022-10-25

**Authors:** Mikaela Coleman, Leonardo Martinez, Grant Theron, Robin Wood, Ben Marais

**Affiliations:** 1WHO Collaborating Centre for Tuberculosis and the Sydney Institute for Infectious Diseases, The University of Sydney, Sydney 2006, Australia; 2Tuberculosis Research Program, Centenary Institute, The University of Sydney, Sydney 2050, Australia; 3Department of Epidemiology, Boston University School of Public Health, Boston, MA 02118, USA; 4DSI-NRF Centre of Excellence for Biomedical Tuberculosis Research, South African Medical Research Council Centre for Tuberculosis Research, Division of Molecular Biology and Human Genetics, Faculty of Medicine and Health Sciences, Stellenbosch University, Cape Town 7602, South Africa; 5Desmond Tutu Health Foundation and Institute of Infectious Disease and Molecular Medicine, University of Cape Town, Cape Town 7700, South Africa

**Keywords:** mycobacterium tuberculosis, TB, transmission, COVID-19, high-incidence

## Abstract

Tuberculosis has affected humankind for thousands of years, but a deeper understanding of its cause and transmission only arose after Robert Koch discovered *Mycobacterium tuberculosis* in 1882. Valuable insight has been gained since, but the accumulation of knowledge has been frustratingly slow and incomplete for a pathogen that remains the number one infectious disease killer on the planet. Contrast that to the rapid progress that has been made in our understanding SARS-CoV-2 (the cause of COVID-19) aerobiology and transmission. In this Review, we discuss important historical and contemporary insights into *M. tuberculosis* transmission. Historical insights describing the principles of aerosol transmission, as well as relevant pathogen, host and environment factors are described. Furthermore, novel insights into asymptomatic and subclinical tuberculosis, and the potential role this may play in population-level transmission is discussed. Progress towards understanding the full spectrum of *M. tuberculosis* transmission in high-burden settings has been hampered by sub-optimal diagnostic tools, limited basic science exploration and inadequate study designs. We propose that, as a tuberculosis field, we must learn from and capitalize on the novel insights and methods that have been developed to investigate SARS-CoV-2 transmission to limit ongoing tuberculosis transmission, which sustains the global pandemic.

## 1. Introduction

Imagine an infectious disease pandemic that has been present in humans for thousands, possibly hundreds of thousands [1] of years and that still afflicts more than 10 million people every year. Between one-third to a half of people that develop the disease do not receive a diagnosis, due to poor disease awareness or health care access, resulting in more than one million deaths from a perfectly treatable disease every year. This is the current global state of the tuberculosis pandemic.

In 18th Century Europe, tuberculosis became the “Robber of Youth” with annual mortality rates of 900 per 100,000 [2]. However, by 1910, mortality rates in New York City and London had declined to 150–300 per 100,000 and by 1950 to less than 50 per 100,000 [3]. Historical declines prior to tuberculosis chemotherapy demonstrate the importance of socio-environmental factors [4,5]. Measuring time spent in crowded poorly ventilated environments frequented by potentially infectious tuberculosis cases reflect socio-environmental risk [6]. In high incidence settings most tuberculosis transmission does not occur in households [7,8,9], but in the general community [10,11].

Among the many reasons for a lack in progress in tuberculosis control is our limited understanding of *M. tuberculosis* transmission. In high incidence settings, the annual risk of *M. tuberculosis* infection, a measure of transmission burden, is estimated to be 5% or higher [12]. In fact, annual risk of infection studies only consider primary infection (not re-infection) and are routinely done in primary school age children in whom the infection signal is easier to measure. However, the annual rate of infection/re-infection is likely to be far higher among adolescents and adults who socialize more actively and do so within groups who are more likely to have infectious tuberculosis. In some high risk settings, such as prisons, annual infection/reinfection rates can exceed 15–20% [13]. Reducing transmission through public health interventions has been difficult, due to a lack of effective tools to identify, measure, and quantify *M. tuberculosis* transmission to guide active case finding and other transmission reduction interventions [14,15]. A deeper understanding of *M. tuberculosis* transmission, using novel methodologies and an open mindset, is necessary to deepen our understanding and inform better targeted interventions.

During the coronavirus disease 2019 (COVID-19) pandemic, collective research was suddenly and overwhelmingly devoted to understanding and interrupting severe acute respiratory syndrome coronavirus 2 (SARS-CoV-2) transmission. Within two years, considerable new insight has been gained, demonstrating the ability offered by new scientific advances, adequate funding and applied minds. The rapidity with which we gained a better understanding of SARS-CoV-2 transmission and the swift implementation of public health measures to try and contain its spread provides an uncomfortable mirror to the slow pace of *M. tuberculosis* transmission research and improved global tuberculosis control (Figure 1). In addition, progress towards ambitious tuberculosis elimination targets were dealt a major blow due to COVID-19 health systems disruption [16,17,18]. It is hoped that the scientific inquiry triggered by COVID-19 will provide renewed insight and resolve to relegate tuberculosis to the history books.

## 2. Historical Insights

### 2.1. Principles of Aerosol Transmission

Airborne transmission requires multiple steps to occur. The first is aerosol generation by an infectious source, which must then be transported through the air to a susceptible host with successful deposition in an anatomical location that facilitates infection (Figure 2). Throughout this journey, the pathogen must retain viability, despite considerable environmental stressors, such as desiccation and ultraviolet light exposure.

*M. tuberculosis* is considered the archetypical example of aerosol transmission through small airborne particles (<5 µm) that remain suspended in the air and ‘contaminate’ poorly ventilated airspaces [19]. Airborne transmission is generally much more difficult to control from an infection control and public health perspective. Coughing and sneezing are the typically identified bodily functions that lead to the production of aerosols, but it is also produced by singing, speaking and breathing [20]. Although particles with culturable bacilli are considered infectious, it is now appreciated that particles with non-culturable bacilli [21,22] may also be infectious [23]. To complete the journey, the infectious particle containing *M. tuberculosis* must reach the small terminal airways deep in the lung that are conducive to infection [24]. This is very different from SARS-CoV-2 that can readily infect mucosal surfaces in the nose and proximal airway.

### 2.2. Spread amongst Close Contacts

People exposed to an infectious individual in their household are at high-risk to acquire *M. tuberculosis* infection and subsequent tuberculosis disease [25]. In Vietnam, household contacts had 2.5 and 6.4 times greater risk of developing ‘any tuberculosis’ and ‘sputum smear-positive disease’ compared to members from the same community without household TB contact [26]. Household contact tracing with use of tuberculosis preventive therapy (TPT) has the greatest individual benefit in vulnerable young children who have the highest risk of developing tuberculosis disease [27,28]. A recent project that evaluated almost 140,000 child household contacts found that 19% of young children with a positive tuberculin skin test (TST) or interferon gamma release assay (IGRA) developed tuberculosis during the subsequent two years [28]. Active case finding and use of TPT among household tuberculosis contacts are advised by WHO and should decrease the time to diagnosis and limit ongoing transmission, but its epidemiological impact has not been demonstrated on a population scale [28,29,30].

Although household exposure is an important risk factor for tuberculosis and provides an opportunity for effective intervention, its contribution to population-level transmission in high incidence settings is thought to be more modest. Molecular studies starting from the 2000’s found that only a small proportion (<20%) of tuberculosis cases were epidemiologically and genetically linked in high incidence settings [8,31,32,33]. For example, a population-based, whole genome sequencing study in Malawi found that only 9.4% of tuberculosis cases could be attributed to known close contact [10]. Tuberculin skin test (TST) surveys with linked household contact tracing have similarly found that a small proportion of individuals with recent *M. tuberculosis* infection have a notified, diagnosed individual with tuberculosis in their household [9,34]. Recent analyses also suggest most transmission to children may occur from outside the household, although this proportion may be slightly lower than adults [35,36]. At a population-level, several factors contribute to *M. tuberculosis* transmission risk [35]. First, the absolute number of community members exposed to a tuberculosis case is thought to be much greater than household members [37]. Therefore, although household contacts of infectious tuberculosis patients are at higher individual-risk of tuberculosis than people without household exposure, its limited population contribution is eclipsed by the much larger number of people exposed in the community (outside the household context). Second, contact saturation in households is likely to occur whereby further transmission opportunities are wasted if multiple exposures of the same individuals occur [38], although the intensity of exposure may also influence disease risk.

These findings suggest that a single intervention, even one with a high individual-level yield such as household contact tracing, may be insufficient to significantly reduce *M. tuberculosis* transmission in settings with a high tuberculosis incidence. Multicomponent interventions that include both household contact tracing and community-based active case finding combined with preventive therapy may be required to interrupt population-level transmission and achieve global goals of tuberculosis elimination [39].

### 2.3. Host-Related Factors in Transmission

In addition to pathogen and environmental factors explored elsewhere, [40] variation in host (i.e., persons that are exposed and at-risk of new *M. tuberculosis* infection and disease; Figure 2) characteristics also influence the risk of ongoing tuberculosis transmission in communities. Demographic characteristics impacting this transmission cycle include age (children who develop tuberculosis are less infectious) [41], sex (men experience higher pulmonary disease rates) [42], smoking status (smoke-related immune suppression, lung damage and chronic cough increase disease risk and possibly also transmission potential) [43,44] and *Bacillus Calmette-Guerin* (BCG) vaccination status (presence of BCG scar correlates with protection against tuberculosis, although this is only consistently observed in young children) [45,46,47]. Co-morbidity such as immune compromise resulting from human immunodeficiency virus (HIV) infection or diabetes [43], prior tuberculosis (identifying the person as vulnerable with potential post-tuberculosis lung scarring) [19,48], the presence of lung cavities (influencing bacterial load) [40,49], number and degree of symptoms (counterintuitively, people with minimal symptoms may produce more infectious aerosols) [50], and length of disease (people with recent disease onset seem most infectious) [51]. All these insights demonstrate the complex interplay of multiple host factors influencing *M. tuberculosis* transmission.

Recent advances in aerosol quantification reveal deeper insights into the mechanics of tuberculosis transmission that extend beyond these host-related factors [20,50,52,53,54]. During tuberculosis disease, host factors determine whether disease is resolved, becomes transmissible, or may lead to ‘superspreading’ [55,56]. One study [50] comparing cough aerosol size produced by patients with drug-susceptible and drug-resistant tuberculosis (DR-TB) found that, aside from sputum bacillary load, host rather than pathogen related factors overwhelmingly drove the number of *M. tuberculosis* cultured from aerosol–a key determinant of transmissive potential.

Cough is a well-established host-factor facilitating tuberculosis aerosol transmission, and is one of the most recognisable tuberculosis symptoms used for passive case detection [19,40,50,57]. However, as a significant proportion of tuberculosis is subclinical and most cases cannot be linked to a known index case [10,32], it is possible that aerosols generated via less telling respiratory activities such as singing, talking and breathing may play a vastly underestimated role in casual transmission [20,52,53,58], Indeed, one South African study sampling respired *M. tuberculosis* from face masks revealed that mask *M. tuberculosis* level correlated more closely with incident infection than sputum bacillary load, or cavitation extent measured by chest radiograph. [53] Crucially, the most common particle sizes in exhaled breath (<5 µM) overlap with the particle sizes known to cause *M. tuberculosis* infection, which are of sufficiently small diameter to deposit in the lower respiratory tract when inhaled [20,24,50]. Further studies examining the characteristics and relative contribution of transmission from asymptomatic and subclinical tuberculosis patients are needed.

### 2.4. Transmission of Drug-Resistant Tuberculosis

The vast majority of DR-TB occurs in people who have never previously had tuberculosis. This indicates that while previous tuberculosis is a risk factor for drug resistance acquisition, transmission is the major driver of its emergence [59] (in a setting where transmission is decreasing or controlled, a higher proportion of DR-TB may be caused by previous drug exposure). In general, DR-TB strain fitness estimates are highly variable [60], because strains are analyzed at different epidemic ages, using different readouts, and in the presence of different putative compensatory mutations. In real life, in contrast to laboratory experiments, more transmissible DR-TB strains are likely to be selected over time and ongoing evolution with extensive transmission have been reported [61]. In a recent study that directly quantified infectiousness by culturing *M. tuberculosis* from captured cough aerosol, no differences in the proportion of DR-TB patients who were aerosol culture-positive were found compared to patients with drug-susceptible tuberculosis [50]. Interestingly neither drug-resistance, the presence of compensatory mutations or strain lineage were associated with cough aerosol culture-positivity.

Importantly, while it remains possible that DR strains are less transmissible than drug-susceptible strains, reductions in strain fitness, if any, are likely to be offset by the greater opportunity such strains have to spread [62]. This opportunity is primarily afforded by delayed initiation of effective treatment. Many tuberculosis cases are never assessed for potential drug resistance and will receive inappropriate treatment in its presence. A lack of universal rapid drug susceptibility testing (DST) in the care cascade creates opportunities for drug-resistant strains to spread and mitigate fitness costs through compensatory mutation in the presence of sub-optimal treatment. Such mutations lead to increasing proportions of transmitted DR-TB in certain settings [63,64]. Furthermore, specific variants may acquire compensatory mutations more readily [65,66], while organism-wide post-genomic effects may also alter fitness. For example, certain *rpoB* mutations may change the structure of the ribosome, affecting the global transcriptome [67]. Furthermore, *Rv0678* variants associated with bedaquiline resistance result in upregulation of a cell wall efflux pump that expels molecules other than bedaquiline [68].

### 2.5. Unusual Routes of Transmission

Airborne transmission of *M. tuberculosis* does not prima facie preclude transmission by more ‘unusual’ routes. Environmental sources in soil [69,70], rivers, wastewater [70,71,72,73], fomites [37,74], dust [75] and cadavers [76] have all been found to harbour viable and infectious *M. tuberculosis* for extended periods of time. As a genus, mycobacteria survive well in soil and water and the robust mycolic acid cell wall contributes to the persistence of mycobacteria in the environment [77]. Most of our understanding of the viability and infectiousness of *M. tuberculosis* found in the environment comes from historical descriptions. These studies [37,75] identified tuberculosis in guinea pigs exposed to infected clinical sputa that had been cultured for up to 90 days with environmental elements (soil, water, dust, fomites). However, human infection resulting from environmental source exposure and the relative contribution of these sources to global infections remain uncertain.

*M. tuberculosis* transmission via topical wound site contamination [78], aerosolisation during surgery, e.g., during tuberculosis abscess drainage or autopsy [79], and ingestion of water contaminated with effluent from tuberculosis sanatoria [80,81] have been observed. In Australia, zookeepers and nearby chimpanzees experienced TST conversion following exposure to a healthy Asian elephant shedding *M. tuberculosis* [82,83]. Gastrointestinal tuberculosis in children caused by swallowing infected milk (mostly *M. bovis* affecting cattle) or infected sputum contribute to the paediatric tuberculosis burden [84], but its relative contribution is small. However, given that the majority of incident tuberculosis cases (~70%) report no known source case in high incidence settings, the potential contribution of unusual modes of transmission warrants further investigation using new tools [10,85]. Undocumented community transmission amongst casual contacts or from people without clinical symptoms may account for most untraced transmission [86,87], but potential aerosolization of environmental particles should also be considered [37,88].

The success of population-wide screen-treat-prevent campaigns in reducing tuberculosis rates in Alaska [89,90], Vietnam [91], and in public health campaigns in Australia, Europe, North America and elsewhere does not suggest a significant role for tuberculosis transmission that is not person-to-person. Nonetheless, there is enough transmission from unknown sources, and some limited ‘real-life’ examples of unusual transmission, to warrant further investigation. Public Health “deep-clean” control measures, used in desperate attempts to contain COVID-19, have not been used for tuberculosis control and do not have any evidence base. However, within 2 years, non-airborne routes of transmission for SARS-CoV-2 have been investigated at length, but comparable investigations for tuberculosis using up-to-date tools and techniques are lacking [92,93,94,95,96] and should include an audit of cleaning and personal protective equipment practices in tuberculosis clinics.

## 3. New Paradigms of Airborne Transmission

### 3.1. Continuum of Droplet-Aerosol Spread

The human lung has been described as an aerosol particle generator [48]. Fluid droplets are generated by all respiratory activities including tidal breathing, speech, singing and coughing. Although the number of particles varies markedly between individuals, within the spectrum of particle sizes, the greater majority are in the 0.5 μm to 5 μm range with a minority of larger (5 μm to 20 μm) size [20,97].

Alveoli and respiratory bronchioles in the peripheral lung alternate between collapse during expiration and opening during inspiration. Surfactant films within these structures stretch and break producing small droplets during inspiration. Some droplets settle by gravity but the remainder pass through the airways with the next expiration [98]. As the exhaled aerosol enters the bronchi and trachea the airstream velocity increases with associated turbulence that interfaces with the fluid lining of the respiratory passages and adds additional larger particles. However, if the airstream is diverted into the nasal passages during a sneeze then individual particles large enough (>50 μm) to be visualized are produced. Pathogen laden aerosols result from the presence of organisms in the fluid at the site of aerosol generation. Culturable *M. tuberculosis* has been isolated predominantly in particles within the 2–5 μm range [24,99]. Particle deposition is predominantly determined by particle size with larger particles >5 μm being deposited in the upper airways where successful infection is less likely, 2–5 μm in the small terminal airways and <2 μm in the alveoli where infection is most likely [100].

As patient generated aerosols move from the warm humid internal environment to the stresses of the external environment and exposed to gravity with the assumption that small particles rapidly become smaller as a consequence of evaporation, while larger particles fall to the ground as determined by Stokes Law. More recent studies however, have shown that patient generated aerosols are exhaled in a high relative humidity cloud with an upward momentum leading to even large particles being projected over many meters [101], with additional spread possible in the presence of air turbulence. In general, airborne *M. tuberculosis* transmission is enabled by proximity or crowding and poor ventilation. *M. tuberculosis* has co-evolved with humans for thousands of years [102] and has adapted to utilise many features of human-generated aerosols to effectively infect new hosts and maintain the chain of infection.

### 3.2. Transmission from Asymptomatic Individuals & Subclinical Tuberculosis

Historically, tuberculosis has been conceptualized as a disease with binary manifestations–‘latent’ tuberculosis infection that is asymptomatic and noninfectious; and active tuberculosis disease that is both symptomatic and infectious. Yet, population surveys reveal that many people diagnosed with microbiologically confirmed tuberculosis display minimal or no symptoms. Such cases are sometimes referred to as the ‘silent man’ phenomenon [103]. Prompting a revaluation of the latent tuberculosis infection–active disease dichotomy, recognition of both incipient tuberculosis (asymptomatic transition from latent infection to early ‘tuberculosis disease’) and ‘subclinical’ tuberculosis (infectious disease with minimal symptoms) as clinical states of active tuberculosis disease is a subject of ongoing debate. Rather than constituting discrete manifestations, a bidirectional continuum of tuberculosis disease–infection has been proposed [104]. The relative contribution of tuberculosis disease in its subclinical manifestation to global transmission remains a subject of intense inquiry. Some estimates suggest as much as 68% of all newly detected cases results from subclinical tuberculosis transmission [105]. Importantly, discrepancies between the contribution of subclinical tuberculosis depend largely on the clinical definitions used in different studies [104,106].

In a South African cohort of adults living with HIV, nearly one quarter of active tuberculosis cases with radiographical evidence of disease reported no tuberculosis-related symptoms [107]. In a non-HIV co-infected population in South Korea, almost 20% of tuberculosis patients had subclinical tuberculosis [108]. Most recently, a review of prevalence survey data from high tuberculosis incidence countries revealed that between 36.1% and 79.7% of prevalent bacteriologically confirmed tuberculosis was diagnosed in individuals that did not report chronic cough or did not have other classic tuberculosis-related symptoms [103,109]. What is clear is that subclinical tuberculosis constitutes a large proportion of tuberculosis cases globally; however, the transmissive potential of subclinical tuberculosis (compared to symptomatic tuberculosis) and its contribution to epidemic spread has been a subject for debate, likely due to the long held paradigm that conflates symptoms with infectiousness.

Some subclinical tuberculosis cases display hallmarks of advanced tuberculosis disease including high bacillary load, smear positivity and extensive lung cavities on chest radiograph [103,105], but many have minimal disease hallmarks despite being positive on microbiological testing. The absence of aerosol generating cough was previously thought to limit the infectious potential of subclinical tuberculosis and coughing probably contributes to infectiousness [19,50,58]. Nonetheless, breathing, singing and talking are all known to produce aerosols of sufficient size and bacillary load for infection [20,52,58,110], The superior lung function of people with subclinical tuberculosis compared with the respiratory deterioration associated with symptomatic disease may enable more robust production of respirable infectious droplet nuclei [50]. In the absence of obvious symptoms, most people will not seek clinical care and, accordingly, will not be diagnosed or commenced on treatment. Added to this is the greater likelihood of people who feel well to maintain high levels of social mixing unlike those with debilitating symptoms. These characteristics combine to describe a manifestation of tuberculosis disease that may be highly transmissible, highly mobile, undetectable in the absence of active screening using sensitive diagnostic equipment and of unknown prevalence in most high-incidence communities. Until such data is available, the reality of subclinical tuberculosis is a powerful reason to advocate for community-wide active tuberculosis case finding in high incidence ‘hot spots’.

### 3.3. Individual-Level Transmission Heterogeneity and ‘Super-Spreaders’

Dynamic transmission models for acute illnesses are often more straightforward than for infections with a longer infection course, like tuberculosis [111]. Transmission heterogeneity is well described in historical studies [112,113]. Individual-level heterogeneity of tuberculosis transmission may occur for a variety of reasons including clinical, environmental, or host-related factors [40]. Using exhaust air funneled from patients in a tuberculosis ward, guinea pigs were then evaluated for acquisition of tuberculosis. A small minority of pulmonary tuberculosis patients caused the vast majority of new tuberculosis infections among the guinea pigs in these studies. These early studies in guinea pigs were later re-created in a laboratory in Lima, Peru. Similarly, exhaust air from a negative-pressure tuberculosis ward was extracted onto ‘detector’ guinea pigs. Guinea pigs were given monthly tuberculin skin tests and were for autopsied if they tested positive [114,115]. These studies found remarkedly similar results as historical studies; 9% of all tuberculosis patients were responsible for 98% of disease detected in guinea pigs.

Several empirical studies have found consistent heterogeneity in the infectiousness of pulmonary tuberculosis patients. A recent study assessing patients using face-mask sampling over a 24-h time period, found wide variability between cough frequency and other tuberculosis-related symptoms, sputum sample production, and *M. tuberculosis* output from face-mask sampling [53]. Other studies using genomic methods or assessing cough frequency have also found high between-patient heterogeneity. For example, in a cough monitoring study, coughing in a 24-h period was below 50 coughs in some patients and above 1000 coughs in others [57]. Similarly, durations of cough have been drastically distinct between groups of patients largely due to high diagnostic delay [116]. Diagnostic delay and longer cough have correlated with increased risk of transmission to a person’s social network suggesting that transmission heterogeneity may be linked to gaps in rapid case detection.

### 3.4. New Aerosol Transmission Insights from the COVID-19 Experience

Rapid advances in our understanding of SARS-CoV-2 transmission are built upon insight gained over many decades through the study of other infectious diseases. Early COVID-19 outbreaks shared typical characteristics of droplet spread, but the virus was soon noted to spread in some settings that could only result from aerosol spread [24]. Importantly, the location of the pathogen in the host, as well as the distinction between airborne and droplet status, has implications for prevention measures (Figure 2). There has been significant debate around the definition of airborne versus droplet transmission. Recognition of aerosol spread is now well accepted, but its actual contribution to overall SARS-CoV-2 transmission remains contentious.

From the beginning of the COVID-19 pandemic, multidisciplinary studies helped to elucidate potential SARS-CoV-2 transmission routes using novel technologies and transmission modelling [117,118,119]. As understanding of transmission evolved, universal indoor mask-wearing (to reduce aerosol and droplet production by infected individuals, as well as inhalation by uninfected individuals) supplanted surface cleaning as the most critical method of prevention. Meta-analysis revealed that mask wearing by both healthy and infected individuals significantly reduced transmission at a population level [120]. Epidemiological data supported the effectiveness of mask wearing measures in public places to curtail COVID-19 spread, irrespective of mask type [121,122,123]. Quality data on the cost-effectiveness and long-term feasibility of mandatory mask-wearing in places with poor ventilation is still limited, and concerns about ‘mask fatigue’ and adherence to public health orders require further investigation, but the effectiveness and short-term viability of such measures during intense transmission waves has been demonstrated [124]. Informing tuberculosis practice, routine N95 mask-wearing by particularly vulnerable individuals to protect themselves, as well as masking of any kind (cloth, surgical or N95) in the less vulnerable, might be employed to similarly successful epidemiological effect. However, such population-level interventions will be hard to sustain, have wider social consequences to consider, and their effectiveness has not been demonstrated in practice [121].

A distinct transmission source that has no equivalent in SARS-CoV-2 infection is the potential of people with tuberculosis infection to experience disease reactivation, which can develop and spread many months or years after the initial exposure event [125,126]. The reality of reactivation (however infrequently this may occur) adds a layer of complexity to tuberculosis transmission not present for SARS-CoV-2, and presents another challenge that tuberculosis research must address. Lessons learned from respiratory virus pandemics, like SARS-CoV-2, provide new insight relevant to the control of tuberculosis transmission, but offer no replacement for dedicated research that considers the unique characteristics of tuberculosis transmission.

## 4. New Paradigms and Tools from Tuberculosis Research

### 4.1. Reducing Community Transmission

Community transmission hot-spots of both tuberculosis and SARS-CoV-2 spread may include public transport [127] churches [128], hospitals [129], homeless shelters [130] and prisons [131]. Other characteristics of community transmission have also been described including settings with poor ventilation and communities with a high degree of social mixing [132]. These findings indicate that methods to increase ventilation in crowded settings for pathogens such as SARS-CoV-2 and tuberculosis needs further consideration and study [133,134,135]. For example, prisons highlight the danger of ignoring known tuberculosis transmission hotspots. In 1903, the medical officer at Clinton Prison, New York reported that prison was a “tubercular death trap” with 40–60% of deaths due to tuberculosis and noted a population risk from discharged prisoners [136]. A recent systematic review estimated the incident rate ratio for tuberculosis in prisons is approximately 10 [13]; incidence rate ratios were higher in low-income settings.

*M. tuberculosis* DNA was recently detected in 18% of South African school classrooms that had air sampled [137], suggesting that significant transmission may be occurring in schools. This study however did not detect live *M. tuberculosis* or measure direct markers of *M. tuberculosis* transmission. If community transmission is to be decreased then environmental monitoring of communal meeting places should be added to existing control programme strategies. Carbon dioxide has been recognised as a measure of adequate ventilation since the 19th Century [138] and monitoring in communal settings allows estimation of per-person ventilation rate, a measure of both crowding and ventilation [139]. Maintaining steady-state levels around 1000 ppm have been recommended by environmental agencies [140]. Carbon dioxide can also be used to measure ventilation and building design and, in health workers, an individual’s level of exposure to carbon dioxide can predict incident tuberculosis infection [141].

### 4.2. Measuring Aerosol Transmission

Distinct to COVID-19, tuberculosis is characterised by obligate airborne transmission [142]. The process of successful transmission requires source generation, expulsion of live *M. tuberculosis* organisms, survival in the environment, inhalation, deposition in the lower lung and establishment of new infection. The historical focus has concentrated on measuring exhaled organisms during coughing with detection by guinea pig [143] or colony forming units [19]. Survival in the environment was demonstrated in airflow transported through air conduits to remote guinea pig cages [113]. Recently, studies utilising facemask collection combined with IS6110 DNA detection have greatly increased the time of collection (8 h) and therefore sampled volume [53]. These mask studies were able to detect subclinical tuberculosis but failed to demonstrate any correlation between cough frequency and quantitative *M. tuberculosis* DNA. Collection of aerosols in the respiratory aerosol collection chamber (RASC), an individual cleanroom [110], demonstrated an airborne time dependent loss of viability [144]. Comparative studies using the RASC and a fluorescent solvatochromic probe enabled comparisons of the *M. tuberculosis* content of coughs, vital capacity and tidal breathing in proven sputum positive tuberculosis cases [20], The finding that exhaled *M. tuberculosis* content was not increased by coughing was consistent with the mask study [53]. Both the mask and RASC studies indicate aerosol collection is no longer restricted to coughing and supports a hypothesis that tuberculosis transmission may be associated with normal breathing, in addition to more violent respiratory activities.

Tuberculosis has long been associated with poverty and it has been postulated that volumes of shared air reflect crowding in poorly ventilated environments and consequently airborne disease transmission risk. Fractional rebreathing can be estimated by carbon dioxide monitoring [139] and can be combined with social mixing data [145]. Total rebreathed air volumes can identify relative tuberculosis transmission risk and help identify transmission hotspots [7,88,146]. Isolation of *M. tuberculosis* DNA in large volume air samples from health care settings [54] and schools [137] is a novel approach to identifying potential tuberculosis risk environments. Recent advances in aerosol collection and identification of *M. tuberculosis* genetic material enhances the identification of tuberculosis transmitters and transmission chains. The isolation of viable aerosolized *M. tuberculosis* presents novel opportunities to explore phenotypic adaptation during transmission, but the pauci-bacillary nature of aerosols will require new developments in single cell microbiology.

### 4.3. Reducing Drug-Resistant Tuberculosis Spread

Generally, initiating drug-resistant tuberculosis patients on effective treatment renders them rapidly non-infectious [147,148]. Importantly, non-infectiousness often precedes non-culturability of *M. tuberculosis* in sputum, in part due to rapidly increasing drug concentrations in droplets arising from droplet evaporation [149,150]. The quickest way of facilitating effective treatment is upfront rapid DST, which is increasingly feasible with new tools like Xpert MTB/XDR [151], however, this requires strengthening the entire drug-resistant tuberculosis care cascade. Furthermore, there are many instances when the drug susceptibility genotype-phenotype associations for different variants are not well understood, limiting the ability to obtain reliable and rapid DST data (for example, phenotypic bedaquiline resistance) [67]. Cough aerosol sampling shows that of the current second-line drugs, inclusion of a fluoroquinolone in the regimen (if the strain is susceptible) is an important independent predictor of reduced infectiousness [50]. Notably, a significant minority of DR-TB patients may remain infectious for a longer period despite being on likely effective treatment. In one study, 16% of patients with rifampicin-resistant tuberculosis were cough aerosol culture-positive after ≥2 weeks of presumed effective treatment [50,152], although it is uncertain if these aerosols were able to establish infection. Unfortunately, while this makes broad generalizations about when DR-TB patients are non-infectious difficult, it should not detract from the importance of urgent effective treatment initiation.

It therefore remains important to always apply standard infection control practices (ventilation, administrative precautions, personal protection), both in patients tested for tuberculosis and in those on DR-TB treatment. Importantly, preventing *M. tuberculosis* transmission can be achieved by measures that work regardless of strain such as masking, personal protective equipment, infection control procedures, ultraviolet light, and increased ventilation. Since most DR-TB transmission is likely to occur before its formal identification and initiation of effective treatment given typical delays in diagnosis, another way to limit its spread may be the use of completely new treatment regimens (so-called pan-tuberculosis regimens) where the chance of pre-existing resistance is very small. This remains hypothetical, but such regimens could also factor in that certain resistance-causing variants [e.g., for pyrazinamide [153]] are associated with greater fitness costs than variants that cause resistance to other drugs. This could reduce the risk of rapid spread when resistance inevitably emerges.

### 4.4. Genomic Transmission Tracking

Recent years have seen rapid advances in whole genome sequencing (WGS) technology, with reductions in cost and incorporation into routine laboratory work flows increasing its viability as an important disease surveillance tool [154]. WGS has proved invaluable during the COVID pandemic to detect the emergence and subsequent geographic spread of new SARS-CoV-2 variants [155]. From a tuberculosis control perspective, WGS has provided unique insight into tuberculosis transmission dynamics within communities [156,157]. The superior resolution of WGS gives us far greater confidence in identifying case clusters and predicting probable transmission pathways than traditional typing methods, such as mycobacterial interspersed repetitive units-variable number tandem repeats MIRU-VNTR) [158]. Furthermore, genomic tracking through population-level genomics provides much greater accuracy and insight into tuberculosis transmission compared to epidemiological mapping of presumed transmission events, which may be subject to incomplete contact information and bias due to the long incubation period of tuberculosis.

Implementation of routine WGS in many low incidence settings allows the identification and measurement of new tuberculosis control targets, but these need consensus definitions and monitoring capacity. An example is the use of ‘zero tuberculosis transmission’ as a tuberculosis elimination definition in low TB incidence countries with high levels of migration [159]. An important benefit from routine WGS for enhanced tuberculosis control in these settings is the ability to guide public health containment responses. Routine WGS can uncover unsuspected local transmission events, focus attention on persistent transmission in particular communities [160] and identify ‘super-spreader’ events where multiple people may become ill following a single high risk exposure or where a specific geographic location serves as a major epidemic amplifier [161,162]. During the COVID pandemic, several countries, including hard-hit countries such as South Africa, began to perform routine WGS. An important open question is whether these countries could successfully pivot their large COVID-19 sequencing capacity to benefit tuberculosis control. The increased use of WGS due to COVID-19 provides an important opportunity for tuberculosis control to perform WGS on as many tuberculosis cases as possible, potentially providing drug-susceptibility testing (permitting rapid effective treatment initiation) and detailed information on tuberculosis transmission at the same time.

The WHO recently recommended 15 actions [163] to accelerate access to genomics for global health, but many hurdles and bottlenecks still hamper wide-spread implementation [164]. Big challenges include the need for pre-culture to perform WGS, the cost and sophistication of the sequencing technology, as well as the availability of high performance computing infrastructure and trained bioinformaticians. The use of optimised pipelines facilitates quality assurance and standardised analysis. The large number of tuberculosis patients and potential for multiple re-infection events occurring in high incidence settings present additional challenges and may complicate transmission inferences. However, the insights gained to date demonstrates that the tuberculosis epidemic in high-incidence settings is sustained by ongoing transmission. In order to impact the tuberculosis epidemic we need better mapping of transmission ‘hot spots’ and an understanding of the factors that sustain them, together with a renewed exploration of interventions that could durably reduce population-level transmission.

In conclusion, despite its long history and the massive ongoing disease burden caused by tuberculosis, much remains to be learnt about *M. tuberculosis* transmission. Rapid advances in our understanding of SARS-CoV-2 transmission are a testament to the power of a shared global will to limit the impact of a devastating pandemic. A similar focus to improve our basic science understanding and implement innovative interventions is necessary to find a pathway towards elimination of the world’s oldest pandemic (Table 1).

## Figures and Tables

**Figure 1 pathogens-11-01228-f001:**
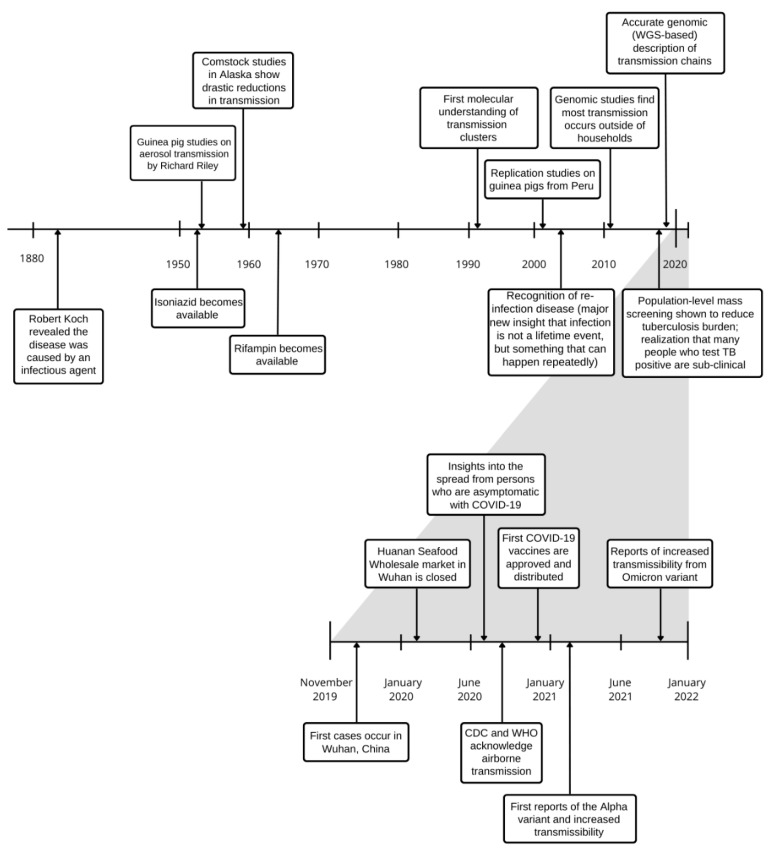
Timeline of major insights into understanding of *Mycobacterium tuberculosis* and SARS-CoV-2 transmission.

**Figure 2 pathogens-11-01228-f002:**
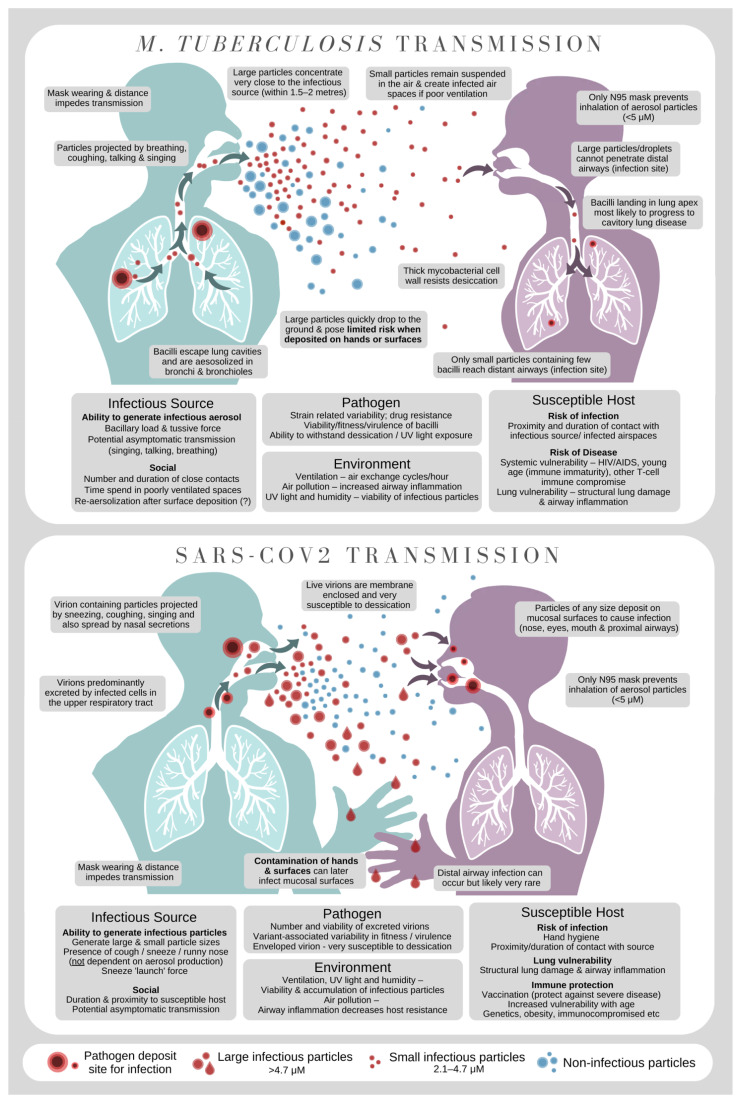
Schematic overview of *Mycobacterium tuberculosis* and SARS-CoV-2 transmission.

**Table 1 pathogens-11-01228-t001:** Key unanswered questions regarding *Mycobacterium tuberculosis* transmission.

Unanswered Questions	Insight from SARS-CoV-2 Research
Are asymptomatic and/or subclinical tuberculosis cases infectious? Furthermore, if so, how infectious are they compared to those with clinically apparent disease and what is the combined contribution to epidemic spread?	Yes; people with asymptomatic SARS-CoV-2 infection shown to be infectious, albeit less infectious than symptomatic cases. However, symptomatic cases contribute a large proportion of population-level transmission
Are there strain related variability in transmission?	Yes; major differences demonstrated in different Variants of Concern (VoCs)
Is there a transmission fitness cost to tuberculosis drug-resistance?	No; little research on drug-resistance, less relevant. Some level of ‘vaccine escape’ associated with VoCs
In which community locations does transmission most commonly occur?	Yes; the vast majority of transmission shown to occur in crowded indoor settings with poor ventilation, including households, pubs and clubs, public transportation (e.g., buses and trains), hospitals, and elderly care settings.
What are the key variables associated with transmission heterogeneity, to close contacts and at population-level?	Yes; wide transmission heterogeneity was demonstrated, with consideration that asymptomatic spread makes a major contribution to population-level spread.
How frequently does superspreading occur and what factors are associated with superspreading?	Yes, a variety of factors have been described-mainly related to the infectiousness of the source case, their participation in large congregate settings and general mobility within the population.
To what extent do potential institutional amplifiers (e.g., prisons, mines, hospitals, churches, schools, etc.) contribute to community-wide tuberculosis transmission?	Not well characterized, but less relevant with extensive population spread.

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
