# Peer review of "Mycobacterium tuberculosis Transmission in High-Incidence Settings—New Paradigms and Insights"

_pathogens, 2022, doi:10.3390/pathogens11111228_

Round 1

Reviewer 1 Report

This is a thoughtful and thorough review and I enjoyed reading it. I appreciate the author’s clear and pithy synthesis of many of these topics and comments on implications for control and prevention of tuberculosis. This review incorporates much recent scholarship on TB transmission which appropriately challenge long-held but sometimes poorly supported notions about TB transmission, but also gives ample space to important contributions of historical, landmark studies. 

Suggested additions for the authors' consideration

Under the heading, ‘New aerosol transmission insights from the COVID-19 experience’ some discussion of how the definitions of airborne vs. droplet transmission have matured in light of the COVID-19 pandemic would be worthwhile. My sense is that the distinction is not particularly useful from a biological standpoint - it requires drawing an arbitrary line about the size of particle on which a pathogen can travel (smaller = airborne and larger = droplet). There has been a lot of work in this area recently (see work of Linsey Marr, for example) that bears noting here. I would consider discussing that where a pathogen lies on the spectrum from drop -> airborne transmission has implications for prevention measures that will be most effective.

Another topic that may fit in this section is insights of transmission model-based methods for measuring transmission heterogeneity of SARS-CoV-2. Lots of literature to source here (e.g., from Max Lau among many others). Notable here is that dynamic transmission models for acute illnesses are often more straightforward than for infections with a longer infection course, like TB. However, there are insights that can be gained from these and similar models (see work by Jonathan Smith on TB transmission heterogeneity in U.S.) and generally still much room for progress in this area.

Section on new tools:

CO2 detection: there has been much interest in increasing ventilation in buildings and other settings to reduce SARS-CoV-2 spread, some literature could be cited here and also some ‘lessons learned’ for using similar measures against TB.

Drug-resistant TB spread: I would liken this to the emergence of new variants of SARS-CoV-2. New variants are similar to drug-resistant Mtb strains in that current treatments/prevention methods are less effective. A salient point here is that in the face of these,  preventing transmission can be achieved by measures that work regardless of strain: masking, PPE, infection control procedures, UV light, increased ventilation. 

Genomic transmission tracking:  important to distinguish between individual-level mapping of transmission events and population-level genomics that aim to understand transmission patterns broadly (I.e. phylogenetic and phylodynamic methods). These are somewhat more difficult to apply in TB than for rapidly-evolving viral pathogens but can still offer insights. 

Minor comments

Figure 1 - should title reflect that SARS-CoV-2 transmission insights are also included?

An aside - neat section on unusual routes of transmission!

“Socially open communities” is not clearly described, please elaborate on the meaning of this so the reader does not need to refer to reference to understand what is meant here.

Author Response

Reviewer 1

General Feedback

This is a thoughtful and thorough review and I enjoyed reading it. I appreciate the author’s clear and pithy synthesis of many of these topics and comments on implications for control and prevention of tuberculosis. This review incorporates much recent scholarship on TB transmission which appropriately challenge long-held but sometimes poorly supported notions about TB transmission, but also gives ample space to important contributions of historical, landmark studies. 

We wish to thank you for this positive review of our manuscript. We have detailed step-by-step actions below to deal with the reviewer’s comments to improve the manuscript.

Comment 1

Under the heading, ‘New aerosol transmission insights from the COVID-19 experience’ some discussion of how the definitions of airborne vs. droplet transmission have matured in light of the COVID-19 pandemic would be worthwhile. My sense is that the distinction is not particularly useful from a biological standpoint - it requires drawing an arbitrary line about the size of particle on which a pathogen can travel (smaller = airborne and larger = droplet). There has been a lot of work in this area recently (see work of Linsey Marr, for example) that bears noting here. I would consider discussing that where a pathogen lies on the spectrum from drop -> airborne transmission has implications for prevention measures that will be most effective.

Thank you for this suggestion. We have now included text discussing the distinction between droplet and airborne transmission. As suggested, we have added text regarding the fact that the location of the pathogen in the host, as well as the distinction between airborne and droplet status, has implications for prevention measures.

Comment 2

Another topic that may fit in this section is insights of transmission model-based methods for measuring transmission heterogeneity of SARS-CoV-2. Lots of literature to source here (e.g., from Max Lau among many others). Notable here is that dynamic transmission models for acute illnesses are often more straightforward than for infections with a longer infection course, like TB. However, there are insights that can be gained from these and similar models (see work by Jonathan Smith on TB transmission heterogeneity in U.S.) and generally still much room for progress in this area.

Thank you for pointing out these references, which we agree are useful for this discussion around transmission heterogeneity. We have now added a citation for SARS-CoV-2 and for tuberculosis, both of which we list below. We have also added text stating that ‘Dynamic transmission models for acute illnesses are often more straightforward than for infections with a longer infection course, like tuberculosis.’

New references:

Smith JP, Oeltmann JE, Hill AN, Tobias JL, Boyd R, Click ES, Finlay A, Mondongo C, Zetola NM, Moonan PK. Characterizing tuberculosis transmission dynamics in high-burden urban and rural settings. Scientific reports. 2022 Apr 26;12(1):1-2.

Lau MS, Grenfell B, Thomas M, Bryan M, Nelson K, Lopman B. Characterizing superspreading events and age-specific infectiousness of SARS-CoV-2 transmission in Georgia, USA. Proceedings of the National Academy of Sciences. 2020 Sep 8;117(36):22430-5.

Comment 3

Section on new tools:

CO2 detection: there has been much interest in increasing ventilation in buildings and other settings to reduce SARS-CoV-2 spread, some literature could be cited here and also some ‘lessons learned’ for using similar measures against TB.

We thank the Reviewer for these comments and agree that adding text on ventilation is useful in this section. Specifically, we now state, ‘Other characteristics of community transmission have also been described including settings with poor ventilation and communities with a high degree of social mixing.[130] These settings indicate that increasing ventilation in crowded, congested settings for pathogens such as SARS-CoV-2 and tuberculosis needs further intervention and study. For example, prisons highlight the danger of ignoring known tuberculosis transmission hotspots.’

We have also added the following citations:

Richardson ET, Morrow CD, Kalil DB, Bekker LG, Wood R. Shared air: a renewed focus on ventilation for the prevention of tuberculosis transmission. PLoS One. 2014 May 7;9(5):e96334.

Somsen GA, van Rijn C, Kooij S, Bem RA, Bonn D. Small droplet aerosols in poorly ventilated spaces and SARS-CoV-2 transmission. The Lancet Respiratory Medicine. 2020 Jul 1;8(7):658-9.

Comment 4

Drug-resistant TB spread: I would liken this to the emergence of new variants of SARS-CoV-2. New variants are similar to drug-resistant Mtb strains in that current treatments/prevention methods are less effective. A salient point here is that in the face of these,  preventing transmission can be achieved by measures that work regardless of strain: masking, PPE, infection control procedures, UV light, increased ventilation. 

We have added text to the manuscript to indicate that preventing M. tuberculosis transmission can be achieved regardless of secondary environmental characteristics such as those listed by the Reviewer. Specifically, we state, ‘Importantly, preventing M. tuberculosis transmission can be achieved by measures that work regardless of strain such as masking, personal protective equipment, infection control procedures, ultraviolet light, and increased ventilation.’

Comment 5

Genomic transmission tracking:  important to distinguish between individual-level mapping of transmission events and population-level genomics that aim to understand transmission patterns broadly (I.e. phylogenetic and phylodynamic methods). These are somewhat more difficult to apply in TB than for rapidly-evolving viral pathogens but can still offer insights. 

We agree and have not clarified further the usefulness of population-level genomics over individual-level mapping of transmission in this section. Specifically, we state, ‘Furthermore, genomic tracking through population-level genomics provides much greater accuracy and insight into tuberculosis transmission compared to individual-level mapping of transmission events, which may be subject to bias due to the long incubation period of tuberculosis.’

Comment 6

Figure 1 - should title reflect that SARS-CoV-2 transmission insights are also included?

Thank you. We have now edited the title of the Figure to reflect SARS-CoV-2 transmission insights as suggested by the Reviewer.

Specifically, the title is now, ‘Figure 1. Timeline of major insights into understanding of Mycobacterium tuberculosis and SARS-CoV-2 transmission’.

Comment 7 

“Socially open communities” is not clearly described, please elaborate on the meaning of this so the reader does not need to refer to reference to understand what is meant here.

We have now edited this phrase to improve clarity. Specifically, we now state, ‘Other characteristics of community transmission have also been described including settings with poor ventilation and communities with a high degree of social mixing.’

Reviewer 2 Report

This review paper by Coleman et al. provides a summary of the historical and contemporary insights into Mycobacterium tuberculosis transmission, correlating with the rapid progress and insights gained by a novel and important pathogen, SARS-CoV-2, that has similarly caused significant worldwide distress.

The paper is well-written with extensive literature cited and summarized within the paper, achieving the objectives set forth as described in the abstract.

I do have some comments that i hope that authors could consider as stated below in improving the paper. 

First, there has also been significant political will to address TB where TB was declared a global health emergency by WHO in 1993, and fight against TB in 2018 to end TB by 2030. Would the authors consider adding comparisons of the implementations contrasting TB with COVID-19, and the impacts on the research activities conducted to address the risk factors associated.

Second, TB as a disease has 2 forms (active and latent), though as stated by the authors could be a continuum, while SARS-CoV-2 is mainly active (perhaps chronic if considering long COVID). 

The paper however did not seem to adequately consider the impact of the latent form of TB disease which might manifest as active subsequently due to a confluence of factors (including age and immunity). This is however seemingly not the case for SARS-CoV-2.   

Page 6-7, it was mentioned that "(in a setting where transmission is decreasing or controlled, a higher proportion of DR-TB will be caused by previous drug exposure)." Can the authors quantify how this might be so and provide references to support this statement?

Page 10, it was stated that "For example, in a cough monitoring study, coughing in a 24-hour period was below 50 in some patients and above 1,000 in others". Could the authors include units of measurement for the 50 and 1,000 values specified?

In addition, I have listed some suggestions with regards to grammar and sentence phrasing which the authors could consider below.

Minor Comments

--------------

Page 2 Introduction

"Between half and one-third of people ..." -> 

Between one-third to half of people ... 

Page 5

"This is very different for SARS-CoV-2 ..." -> 

This is very different from SARS-CoV-2 ...

Page 5

"A recent project evaluated almost 140,000 child household contacts finding that 19% ..." -> 

A recent project that evaluated almost 140,000 child household contacts found that 19% ...

Page 5

"Active case finding and use of TPT among household tuberculosis contacts is advised by WHO ... "->

Active case finding and use of TPT among household tuberculosis contacts are advised by WHO

Page 5

"... but this has not been demonstrated ..." ->

... but these have not been demonstrated ...

Page 5

"Therefore, although household contacts of infectious tuberculosis patients are at higher individual-risk of tuberculosis 

than people without household exposure, its limited population contribution is driven by the much larger number of people 

exposed in the community (outside the household context). " ->

Therefore, although household contacts of infectious tuberculosis patients are at higher individual-risk of tuberculosis 

than people without household exposure, its limited population contribution is eclipsed by the much larger number of people 

exposed in the community (outside the household context).

Page 6

" ... and degree of symptoms (counterintuitively people with minimal symptoms may produce more infectious aerosols) ..." ->

... and degree of symptoms (counterintuitively, people with minimal symptoms may produce more infectious aerosols) ...

Page 6

" ... comparing cough aerosol size produced by patients with drug-susceptible and DR-TB found ... " ->

The abbreviation DR-TB is only defined later in the section "Transmission of drug-resistant tuberculosis"

Page 6

"The vast majority of drug-resistant tuberculosis (DR-TB) occurs in people who have never had tuberculosis before previous tuberculosis is a risk factor ..." ->

The vast majority of drug-resistant tuberculosis (DR-TB) occurs in people who never had tuberculosis before previous tuberculosis disease is a risk factor ...

Page 7

"For example, certain rpoB mutations may change the structure of the ribosome, effecting the global transcriptome." ->

For example, certain rpoB mutations may change the structure of the ribosome, affecting the global transcriptome

Page 9

"Importantly, there large dis-crepancies between the contribution of subclinical tuberculosis ..." ->

Importantly, the large discrepancies between the contribution of subclinical tuberculosis ...

Page 9

"In the absence of obvious symptoms, most people will seek clinical care and ..." ->

In the absence of obvious symptoms, most people will not seek clinical care and ...

Page 10

"Rapid advances in our understanding of SARS-CoV-2 transmission built upon in-sight gained over many decades through the study of other infectious diseases."->

Rapid advances in our understanding of SARS-CoV-2 transmission are built upon in-sight gained over many decades through the study of other infectious diseases.

Page 10

"... but it was soon noted to spread in some settings could only result from aerosol spread." ->

... but it was soon noted to spread in some settings that could only result from aerosol spread.

Page 12

"Unfortunately, this makes broad generalizations about when DR-TB patients are non-infectious difficult ..." ->

Unfortunately, while this makes broad generalizations about when DR-TB patients are non-infectious difficult ...

Box

Point 1

TB: "... and what is there combined contribution to epidemic spread?" ->

... and what is the combined contribution to epidemic spread?

SARS-CoV-2: "However, contribute a large proportion of population-level transmission" ->

... However, symptomatic cases contribute a large proportion of population-level transmission

Point  5

SARS-CoV-2: "... with consideration that asymptomatic spread is makes a major contribution to population-level spread." ->

with consideration that asymptomatic spread makes a major contribution to population-level spread

Author Response

Reviewer 2

General Feedback

This review paper by Coleman et al. provides a summary of the historical and contemporary insights into Mycobacterium tuberculosis transmission, correlating with the rapid progress and insights gained by a novel and important pathogen, SARS-CoV-2, that has similarly caused significant worldwide distress.

The paper is well-written with extensive literature cited and summarized within the paper, achieving the objectives set forth as described in the abstract.

I do have some comments that I hope that authors could consider as stated below in improving the paper. 

            Truly grateful for this generous feedback.

Comment 1

First, there has also been significant political will to address TB where TB was declared a global health emergency by WHO in 1993, and fight against TB in 2018 to end TB by 2030. Would the authors consider adding comparisons of the implementations contrasting TB with COVID-19, and the impacts on the research activities conducted to address the risk factors associated.

Given the incredible breadth of subject matter that could be covered in a comparison between TB and SARS-CoV-2, we elected to narrow the focus of the review towards new insights in  M. tuberculosis transmission, with particular emphasis upon the mechanical aspects of transmission, given recent progress in this area. We certainly (and regretfully) have omitted discussion around the important WHO strategies designed to interrupt transmission of both diseases. Such a comparison deserves thorough exploration, which we feel cannot be done justice within the scope of this review.

 Comment 2

Second, TB as a disease has 2 forms (active and latent), though as stated by the authors could be a continuum, while SARS-CoV-2 is mainly active (perhaps chronic if considering long COVID). 

The paper however did not seem to adequately consider the impact of the latent form of TB disease which might manifest as active subsequently due to a confluence of factors (including age and immunity). This is however seemingly not the case for SARS-CoV-2.   

This is an important point of distinction between TB and SARS-CoV-2, and we have sought to include this comparison on pages 11-12 with the following text addition:

A distinct transmission source that has no equivalent in SARS-CoV-2 infection is the potential of people with tuberculosis infection to experience disease reactivation, which can renew disease spread many months or years after the initial exposure event and in settings otherwise without known community transmission.[125,126]  The reality of reactivation (however infrequently this may occur) adds a layer of complexity to tuberculosis transmission not present for SARS-CoV-2, and presents a challenge that tuberculosis research must address in its own right. Lessons learned from experiences with other respiratory epidemics may be highly relevant to the control of tuberculosis transmission, but these insights are no replacement for dedicated research heedful of the unique characteristics of tuberculosis transmission.”  

 Comment 3

Page 6-7, it was mentioned that “(in a setting where transmission is decreasing or controlled, a higher proportion of DR-TB will be caused by previous drug exposure).” Can the authors quantify how this might be so and provide references to support this statement?

The sentence was qualified to read “in a setting where transmission is decreasing or controlled, a higher proportion of DR-TB may be caused by previous drug exposure”.

 Comment 4

Page 10, it was stated that "For example, in a cough monitoring study, coughing in a 24-hour period was below 50 in some patients and above 1,000 in others". Could the authors include units of measurement for the 50 and 1,000 values specified?

            As suggested, the unit of measurement for these values has been included in the text to read;

For example, in a cough monitoring study, coughing in a 24-hour period was below 50 coughs in some patients and above 1,000 coughs in others.

 Minor comments

In addition, I have listed some suggestions with regards to grammar and sentence phrasing which the authors could consider below.

Thank you! All changes have been made in accordance with these grammar and sentence phrasing comments.

Reviewer 3 Report

This article is well-written with sufficient and concise details on tuberculosis research and linking it to the Covid-19 pandemic era. I support the publication of this review article.

Author Response

Reviewer 3

General Feedback

This article is well-written with sufficient and concise details on tuberculosis research and linking it to the Covid-19 pandemic era. I support the publication of this review article.

We wish to thank you for spending time to read and give feedback on our manuscript.